# Decoding the language of microbiomes using word-embedding techniques, and applications in inflammatory bowel disease

**Christine A. Tataru** [1]*, **Maude M. David** [1,2]*

**1** Department of Microbiology, Oregon State University, Corvallis, Oregon, United States of America,
**2** Department of Pharmaceutical Sciences, Oregon State University, Corvallis, Oregon, United States of America

* tataruc@oregonstate.edu (CAT); maude.david@oregonstate.edu (MMD)

## Abstract

Microbiomes are complex ecological systems that play crucial roles in understanding natural phenomena from human disease to climate change. Especially in human gut microbiome studies, where collecting clinical samples can be arduous, the number of taxa considered in any one study often exceeds the number of samples ten to one hundred-fold. This discrepancy decreases the power of studies to identify meaningful differences between samples, increases the likelihood of false positive results, and subsequently limits reproducibility. Despite the vast collections of microbiome data already available, biome-specific patterns of microbial structure are not currently leveraged to inform studies. Here, we derive microbiome-level properties by applying an embedding algorithm to quantify taxon co-occurrence patterns in over 18,000 samples from the American Gut Project (AGP) microbiome crowdsourcing effort. We then compare the predictive power of models trained using properties, normalized taxonomic count data, and another commonly used dimensionality reduction method, Principal Component Analysis in categorizing samples from individuals with inflammatory bowel disease (IBD) and healthy controls. We show that predictive models trained using property data are the most accurate, robust, and generalizable, and that property-based models can be trained on one dataset and deployed on another with positive results. Furthermore, we find that properties correlate significantly with known metabolic pathways. Using these properties, we are able to extract known and new bacterial metabolic pathways associated with inflammatory bowel disease across two completely independent studies. By providing a set of pre-trained embeddings, we allow any V4 16S amplicon study to apply the publicly informed properties to increase the statistical power, reproducibility, and generalizability of analysis.

## Author summary

The gut microbiome in humans has been implicated in a spectrum of diseases including inflammatory bowel disease, anxiety, depression, and Parkinson's Disease, but thus far the associations between qualities of the gut microbiome and host symptoms are often not

---

**Data Availability Statement:** All raw data files are already available, as this research did not generate new raw data files. American Gut Project data are available from the NCBI database (accession

---

number PRJEB11419). The Halfvarson data are available from the NCBI database (accession number PRJEB18471). Code can be found at https://github.com/MaudeDavidLab/gut_microbiome_embeddings and data files at http://files.cgrb.oregonstate.edu/David_Lab/microbiome_embeddings/.

**Funding:** This work was funded by Oregon State University start-up funds of M.D. The funders had no role in study design, data collection and analysis, decision to publish, or preparation of the manuscript.

**Competing interests:** The authors have declared that no competing interests exist.

consistent across datasets. This may be because individual microbiome studies generally contain relatively small sample sizes, because some microbes are present in some populations but not others, and because microbial metabolism is dependent on the environmental context at hand. At the same time, there is a plethora of publicly accessible data describing the gut microbiome compositions of thousands of individuals in addition to their disease status, dietary habits, and lifestyle choices. We have employed a word embedding algorithm to map gut microbes from massive public datasets to vectors of real numbers which then represent relationships between microbes, or microbe-microbe co-occurrence patterns. We then use this mapping to learn more about what the gut microbiome of individuals with inflammatory bowel disease looks like, and find that mapping microbes to their vectors allows us to generalize results from one population to another more accurately.

## Introduction

### Microbial survey studies

Recent findings suggest that resident microbiomes of the human anatomy influence our bodies and minds in ways we have only just begun to understand. Microbiomes have been implicated in the development of diseases of nearly all types, both acute and chronic, infectious and systemic. The vaginal microbiome has been implicated in preterm birth [1], the skin microbiome in acne [2] and eczema [3], and the gut microbiome in a spectrum of diseases including inflammatory bowel disease (IBD) [4–6,6–9], anxiety [10–12], major depressive disorder [13–15], autism [16–20], and Parkinson's Disease [21–23].

To analyze microbiome compositions, current technology sequences various hypervariable regions of the 16S rRNA gene, which acts as an accessible taxonomic tag to measure the abundances of taxa in a community. Studies using this 16S survey technique have reported incredibly diverse collections of microbes in several systems. Multiple individuals studies, along with consortium studies like the American Gut Project (AGP) [24] and the Human Microbiome Project [25], have invested colossal effort to document that diversity by creating publicly available reference repositories. Amongst these are repositories of stool-associated microbiota that have furthered our understanding of the role of the microbiome in several diseases, especially inflammatory bowel disease (IBD) [4]

Though these and other studies have presented highly relevant findings, 16S microbiome survey studies in general tend to suffer from lack of power due to necessarily small sample sizes. Due to logistical restrictions, even the largest microbiome studies only include roughly as many samples as taxa, and samples may outnumber taxa ten to one hundred-fold [24,25]. Studies with low sample-to-variable ratios risk being underpowered and reporting false positives, especially when effect sizes are estimated to be small [26–28].

In addition, in biological contexts, the presence and function of each microbe is deeply dependent on the characteristics of its surrounding neighbors. Differences in microbial function also occur as genes are turned on or off as appropriate for that microbe's environment at any given time. For instance, Belenguer et al. show that *Roseburia* strain A2-183 is unable to use lactate as a carbon source except in the presence of *Bacteroides adolecentis* [29]. Because of functional dependence, findings of differential abundance or function of a single species must be considered within its wider context of associated species and environmental factors [30]. To address this problem, it is often appropriate to base analysis on the log-ratios of pairs of taxa, which allows identification of differences in the relationship of taxa pairs between groups [31].

Lastly, despite the vast amount of publicly available 16S microbiome survey data, current studies design their data collection and perform their analysis independently, without leveraging the information available via massive sequencing projects such as the Human Microbiome Project [25] or American Gut Project [24].

## Current methods for dimensionality reduction

Dimensionality reduction techniques, both feature selection and feature transformation, may be performed to reduce microbiome survey study data to a manageable size. Feature selection techniques include filtering to consider only the common or very rare, however this approach may filter potentially valuable data. Feature transformation techniques include binning taxa by their annotations (e.g. all taxa that share a family are analyzed as one unit) [32,33] and clustering taxa based on the similarity of their 16S rRNA gene, which has been used as a proxy for evolutionary relatedness [34,35]. The goal of dimensionality reduction is to remove variables that add noise while preserving those that contain signal with respect to host phenotype. While all of these methodologies have been used to produce interesting and crucial discoveries in microbiome surveys, they all rely on presupposed assumptions about which variables will be most informative of host phenotype.

Another form of feature transformation, ordination, may instead be used to identify broad patterns in microbiome compositions between samples. Samples, each represented by a vector of taxa, can be projected into a lower dimensional space using a wide array of ordination techniques including principal correspondence analysis (PCoA) [36], multidimensional scaling [37], and non-metric multidimensional scaling [38]. Broadly used, ordination has played a critical role in associating microbial structure or diversity with specific features or phenotypes of interest, but has also proven to be overly sensitive to normalization and study bias (e.g. technological noise, DNA preparation protocol, sequencing error) [39].

Patterns in 16S sequence relatedness have been used for feature transformation as well. Woloszynek et al. represent each 16S sequence by the set of k-length nucleotide sequences (k-mers) it includes, and embed those k-mers to create a vector representation of each sequence [40]. Finally, we may use taxon counts to estimate parameters for an underlying distribution from which taxa are drawn. Sankaran et al. model taxa as units drawn from a latent Dirichlet multinomial mixture distribution [30].

While compelling, the dimensionality reduction methods described above do not consider taxonomic relationships within a biological context, or make use of information already available from previous datasets.

## Current study overview

Navigating the highly related and very large microbiome space can be done by using the information encompassed in publicly available datasets to inform novel dimensionality reduction methods. The goal of this project is to create an unbiased method to project taxonomic data into a lower dimensional space that represents properties of taxa. Properties are based on taxa's relationships with each other and their environment, are learned from public datasets, and are re-usable for past and future studies. In this context, a property is a pattern that underlies co-occurrences between taxa. The lower dimensional space is learned from public datasets using an embedding algorithm, and allows the integration of patterns from massive datasets into specialized studies to increase generalizability and statistical power. A graphical representation of workflow can be found in Fig 1, and a more detailed explanation can be found in the workflow section of Methods.

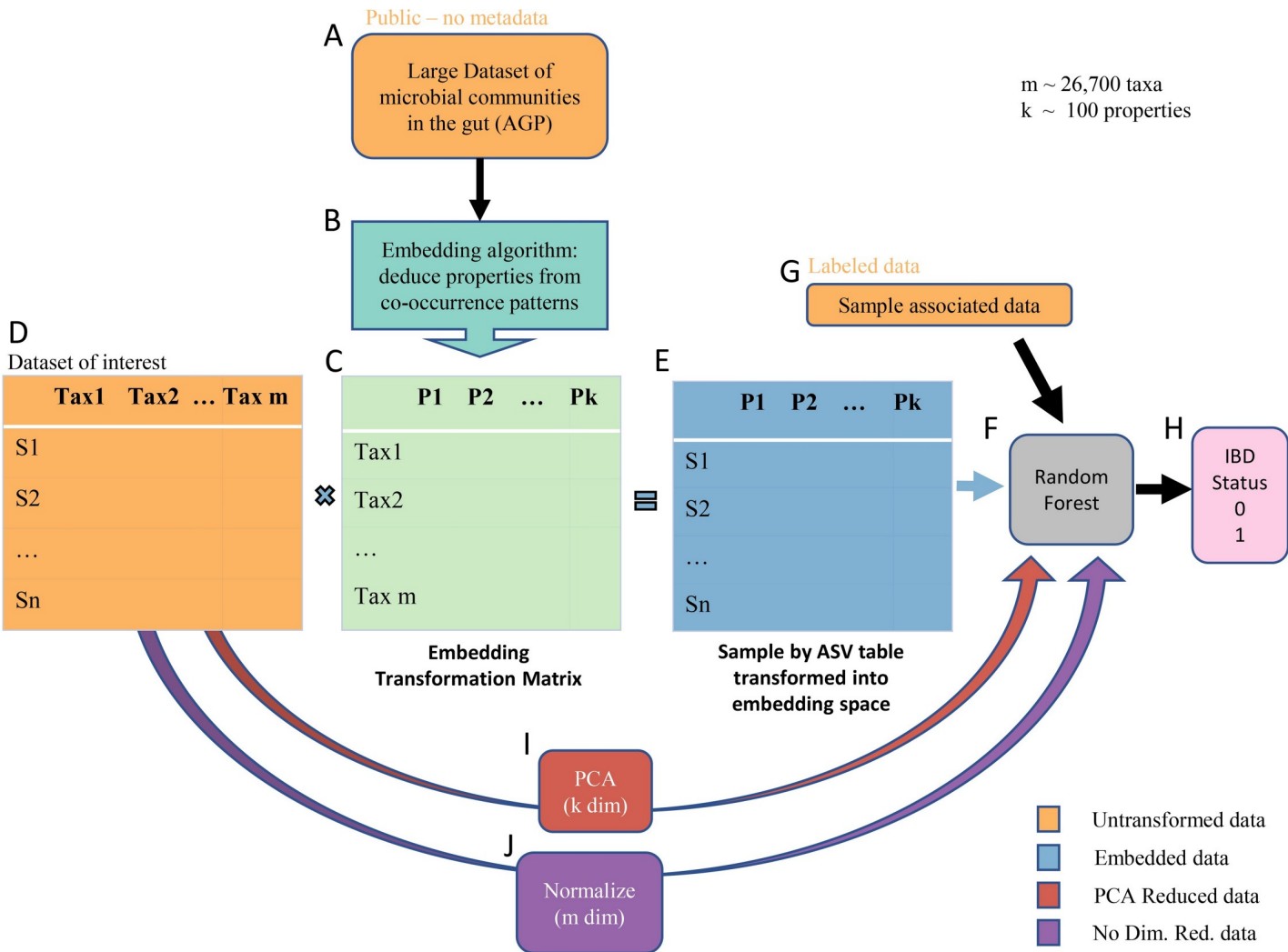

**Fig 1. Workflow of data transformation to prediction of host phenotype.** First, taxon-taxon co-occurrence (binary) data from the American Gut Project (A) are input into the GloVe embedding algorithm (B) to produce a taxon (Amplicon Sequence Variant or ASV) by property transformation matrix (C). Then, we take the dot product between a sample by ASV table of interest (D) and the transformation matrix (C) to project that table into embedding space (E). The ASVs in the columns of D and those in the rows of C must match exactly. This table is used to train a random forest model (F) along with sample associated lifestyle and dietary information (G, optional) to predict the IBD status of the host (H). As points of comparison, random forest models are also built without embedding, after transforming the same sample by ASV table (D) using PCA (I) and normalizing (J).

## Results

By applying the GloVe algorithm to 18,480 gut microbiome samples from the American Gut Project, we construct a 26726 by 26726 ASV co-occurrence matrix, and subsequently a 26726 ASV by 100 property embedding transformation matrix, which can be used to project any sample by ASV table into embedding space. We then test the performance of models trained on taxonomic counts, embedded data, and pca transformed data across combinations of training/testing datasets: 1. Models trained and tested on the same dataset used to construct embeddings (AGP) 2. Models trained and tested on an independent IBD dataset (Halfvarson) and 3. Models trained on one dataset (AGP) and tested on two independent sets (Halfvarson and Schirmer). We find that models trained on embedded data perform as well or better than models trained on ASV count data despite using 250 fold fewer variables, and far outperformed

models trained on PCA reduced data. We find that metabolic functional capacity exhibits significant correlation with property structure, and suggest that metabolic pathways such as steroid degradation, lipopolysaccharide biosynthesis, and various types of glycan biosynthesis may be relevant to IBD.

## Model performance

In order to determine the value of the set of embedding produced by GloVe, we tested the performance of classifiers built using GloVe embedded, PCA transformed, and non-embedded normalized count data. We evaluated two main performance metrics in predicting the IBD status of the host: area under the receiver operating curve (AUROC) and area under the precision-recall curve (AUPR).

## Pick optimal number of properties to define a community

We found random forest classifiers trained using GloVe embedded data produce a significantly higher average area under the Receiver Operating Curve (AUROC) across all choices of hyperparameters and number of dimensions (Fig 2) than non-embedded data and PCA-embedded data ($p \ll 0.05$, rank sum test). Notably, embedded data consistently produces better results with far fewer features than taxonomic counts. The use of fewer features makes the model less likely to overfit the data and more likely to be reproducible. We run all future tests using 100 properties, as models trained with 100 properties show the most consistently high performance and small variance across all hyperparameter choices.

## Models built with embedded data perform better on a held out test set

We then train three separate models on the training portion of the AGP dataset, and test each model on a held out portion of the same dataset that has been used neither for model nor embedding training (Fig 3 panel A). Each model uses a different data input type, GloVe

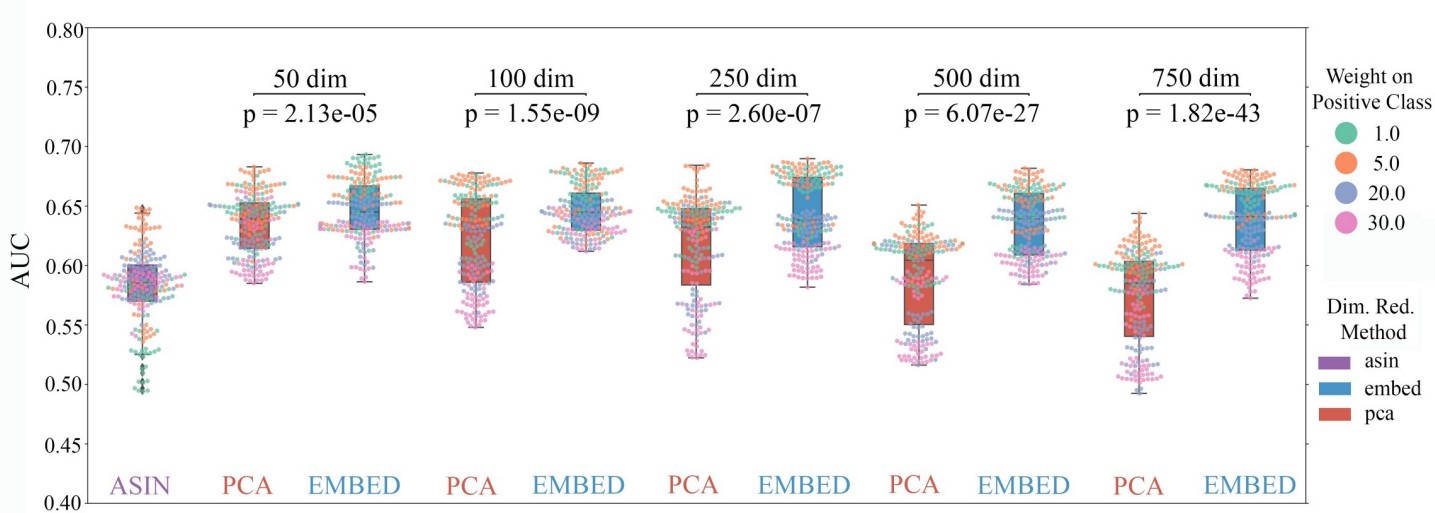

**Fig 2. Transforming ASV tables into GloVe embedding space before training a model produces more accurate host phenotype predictions (IBD vs. healthy control) and makes models more robust to hyperparameter choice.** Each point represents a triplet of choices for number of trees, depth of each tree, and weight on a positive prediction of IBD in a random forest model. Each model was trained on the data input type indicated by color (Normalized, non-embedded counts is purple, PCA embedded data is pink, and GloVe embedded data is blue). Models trained on GloVe embedded data produce higher ROC AUCs with less variance across hyperparameter choice.

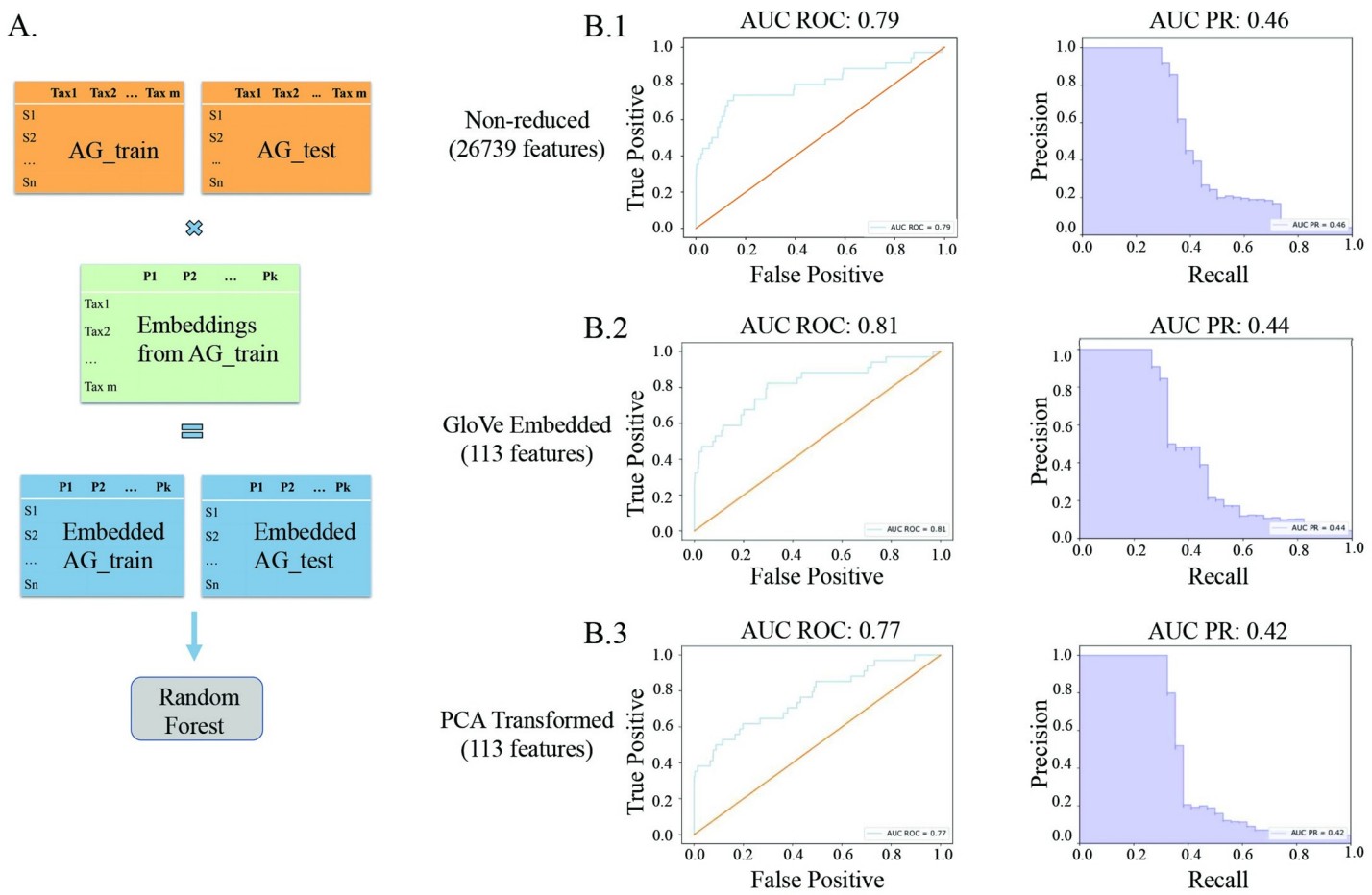

**Fig 3.** Embeddings trained on American Gut training set, model trained on American Gut training set, model tested on American Gut held out test set (A). Models trained on GloVe embedded data have higher ROC AUC but slightly lower Precision-Recall AUC on a held out test set (B).

embedded, PCA-transformed, or non-embedded normalized ASV counts, and has hyperparameters optimized using cross-validation over the training set. We see comparable performance between the classifier using GloVe embedded data and the other two methods (Fig 3 panel B). The model with non-embedded data, which uses 26,739 features, has an area under the Receiver Operating Curve (AUROC) of 0.79 and an area under the Precision-Recall curve (AUPR) of 0.46 (Fig 3, panel B.1). In contrast, the model using GloVe embedded data, which uses only 113 features, has a higher AUROC of 0.81 but slightly lower AUPR of 0.44 (Fig 3 panel B.2). A 200-fold decrease in number of features used results in little change in relevant performance metrics. In comparison, the model using PCA-transformed data with 113 features performs only slightly worse, with an AUROC of 0.77 and an AUPR of 0.42 (Fig 3 panel B.3)

## Properties are generalizable to independent stool-associated datasets

We find that GloVe embedded data generalizes to a completely independent datasets, and improves performance over normalized count data regardless of the number of training samples used. Using data from Halfvarson et al. [8], we train random forest classifiers on gut microbiome data to differentiate between IBD vs. healthy control (Fig 4A). Again, we train classifiers using normalized count data, PCA-embedded data, and GloVe embedded data, and optimize over hyperparameters using cross-validation for each model independently. To test

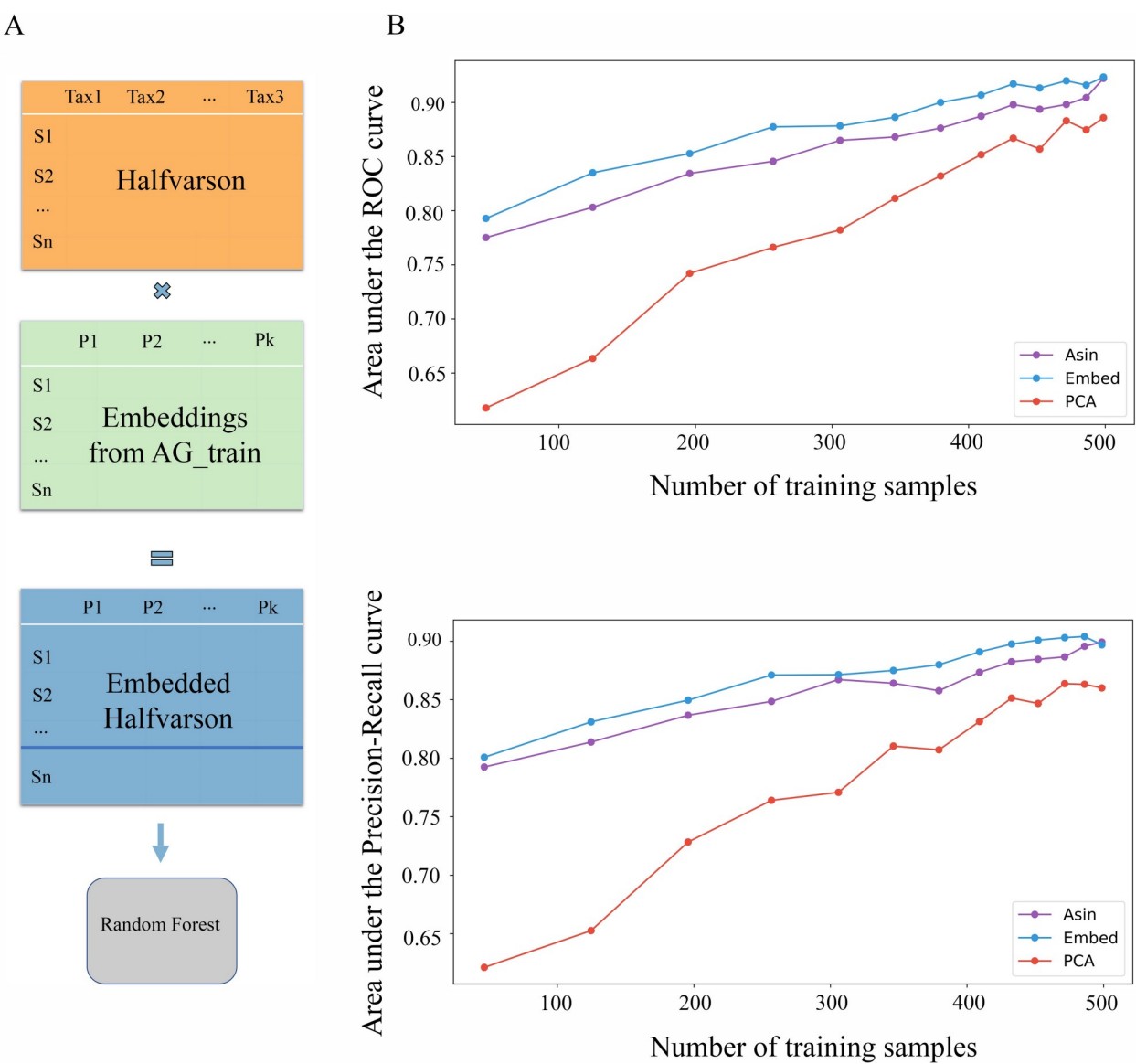

**Fig 4.** Embeddings were trained on American Gut data, and the predictive models were trained and tested on Halfvarson dataset (A). Transforming microbiome data into GloVe embedding space (100 features) prior to model training produces more accurate models than using ASVs (26,251 features) (B).

the effect of training set size on performance outcomes, we train models using from 50 to 450 samples in the training set, and the rest in the test set. In this dataset, we have 564 samples from 118 patients and 26,251 Amplicon Sequence Variances (ASVs). We do not include any associated metadata; predictions are made solely based off of the microbiome compositions. Additionally, results are averaged over train/test splits using 100 different random seeds.

It is important to note that the transformation matrix that puts the query dataset into embedding space is trained exclusively on American Gut Project data, and is therefore completely independent of the query dataset. Despite the fact that properties were learned using the American Gut data dataset exclusively, we see embedding model performance is consistently higher than the ASV-based model on the independent Halfvarson dataset (Fig 4B). ASVs from the query datasets were considered to match an embedding ASV if they were more

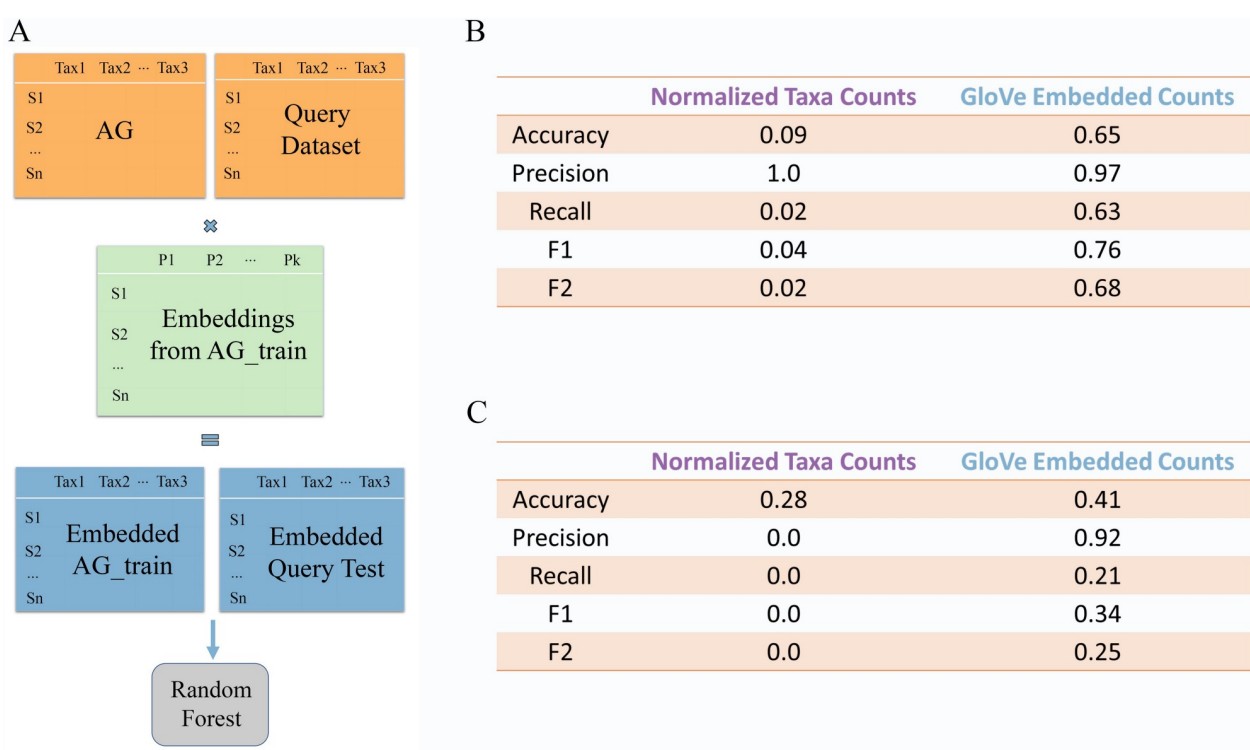

**Fig 5.** Two models, one embedding-based and one ASV-based, were trained on American Gut data and tested on two independent query datasets (A). Embedding-based models outperform ASV-based models significantly when testing on Halfvarson dataset (B) and Schirmer dataset (C).

than 99% identical. Largely equivalent results obtained using a similarity threshold of 97% and 100% are shown in S1 Fig. The patterns learned by the GloVe algorithm from the American Gut data generalize to improve classification performance on an independent dataset. Theoretically, classification accuracy of any host phenotype relating to the gut microbiome could be bolstered by first embedding the input data before model training.

## Models that use properties are generalizable to independent datasets

In the above experiments, all models were trained on the same datasets they were tested on, using cross-validation and a held-out test sets. Now, we trained a model on the American Gut data and tested it on two independent datasets, the Halfvarson data described above and a second independent IBD focused dataset created by Schirmer et al. [9] (Fig 5A). More so than a hold-out test set, this allows us to test the feasibility of deploying a model for diagnosis and monitoring of IBD. Two models were trained on American Gut data, one using normalized ASV counts and the other ASV counts embedded in property space. In this case, only microbiome data and no sample-associated data was included. Hyperparameters that gave the highest area under the precision-recall curve during 3 fold cross validation on American Gut data were selected, and the trained model was directly applied to the independent datasets without re-tuning hyperparameters or decision thresholds. For the Halfvarson test set, both models trained on ASV counts and embedded data had a precision very near 1, meaning that a positive IBD prediction was correct the vast majority of the time. However, the model trained on ASV counts had a recall of 0.02, meaning that only 2% of the samples from patients with IBD were positively identified. In contrast, the model trained on embedded data recovered 63% of samples from patients with IBD (Fig 5B). On the Schirmer dataset, the ASV model predicted only healthy samples, and so had precision and recall scores of 0, while the embedded model had a

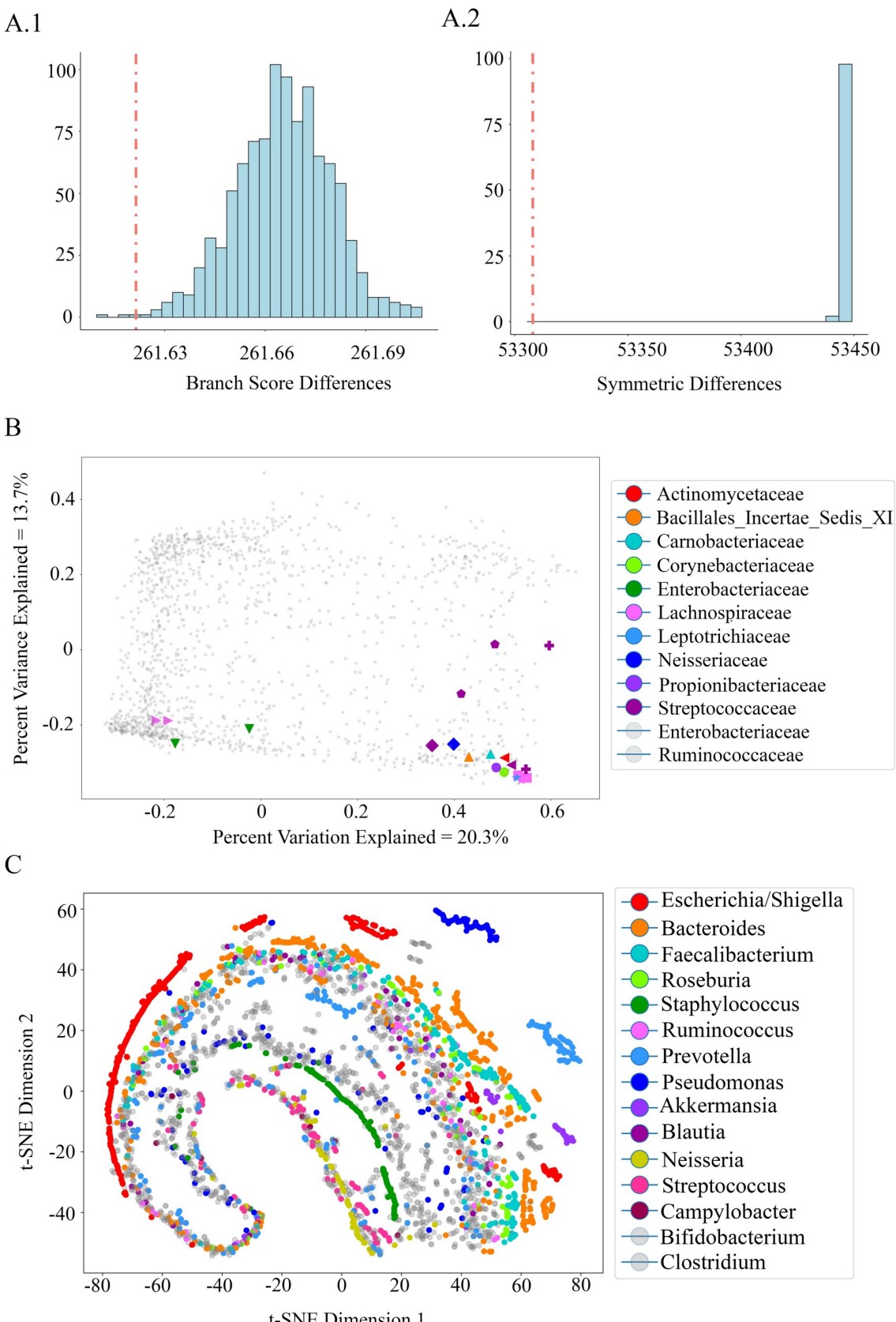

**Fig 6. Relationship between embedding space and phylogeny. A**: Hierarchical clustering of ASVs using similarity between property vectors matches phylogenetic tree topology. Histograms (light blue) show branch score distances (A1) and symmetric distances (A2) between permuted phylogenetic trees and hierarchically clustered trees based on property similarity. The dotted red line shows the distance between the true phylogenetic tree and the property-based tree. **B**: 2D PCoA ordination using cosine distance of 1500 randomly selected ASVs in embedding space. 10 pairs of ASVs were selected based on their high spearman correlation in the American Gut ASV table across all samples(r > 0.7). Each pair is denoted by shape. Color shows the family classification of the selected pairs, and all other ASVs are shown in grey. **C:** t-SNE ordination using cosine distance of 5000 randomly selected ASVs, colored by genus. 3 genera highlighted in the dashed circle exhibit similar co-occurrence patterns.

precision of 92% and a recall of 21% (Fig 5C). Equivalent results obtained using similarity thresholds of 97% and 100% are shown in S2 Fig. While the model trained on ASV counts was in no way generalizable to other datasets, the model trained on data in property space was able to make accurate predictions on a completely independent dataset. This finding demonstrates that in this case, models built from embedded data can generalize to outside data while models built from ASV abundance information cannot.

## Property relationship with phylogeny

ASVs close together in embedding space have similar co-occurrence patterns by construction. We expect that phylogenetically close taxa are more likely to fill the same ecological niches than are unrelated taxa. We therefore expect a slight but not extreme relationship between phylogeny and property metrics.

We built two trees with ASV leaves, one clustered by phylogeny and the other clustered by property values. We then calculated the branch score difference and symmetric distance between the two, and compared them to null distributions created by permuting the leaves of the phylogenetic tree 1000 times. We find that the tree built using property values of ASVs is significantly similar (p = 0.003) to the phylogenetic tree, supporting a phylogenetic component in the determination of co-occurrence patterns captured by properties (Fig 6A1 and 6A2). The large magnitude of the branch score difference suggests that phylogeny is not the main driving factor of the described co-occurrence patterns.

We also find that ASV pairs with the highest spearman correlations of counts across samples are generally close in embedding space, as expected. Of the 10 annotated ASV pairs with highest spearman correlation, we find 5 share a family while 5 do not (Fig 6B). To further this analysis, we also identified 3 pairs of ASVs that were close together in embedding space, but *did not strongly* directly correlate in raw counts. These ASVs are more likely to share a similar biological function and potentially even compete. The pairs include *Bacteroides vulgatus* and *Escherichia coli*, *Bacteroides uniformis* and *Faecalibacterium prausnitzii*, and *Faecalibacterium prausnitzii* and *Eubacterium rectale*.

Lastly, to visualize embedding space in the context of phylogeny, we plot a 2D t-SNE ordination of embedding space with ASVs colored by genus (Fig 6C). 5000 ASVs were randomly selected to represent the space, and 13 genera were selected to be colored. In particular, the three genera Staphylococcus (green), Streptococcus (pink), and Neisseria (yellow) appear close together with similar trajectories in this space and do not appear elsewhere, implying that these genera may either interact directly or share some similarities. Moreover, a PCoA ordination using correlation of normalized counts (not embedding space) shows that these 3 genera do often occur in direct correlation with one another, implying the potential for microbes from these genera to interact synergistically (S3 Fig).

## Property relationship with metabolic capacity

We chose to preserve taxa co-occurrence patterns in embedding space because we hypothesize that those patterns are driven by taxa functionality in an environment. As such, we evaluate

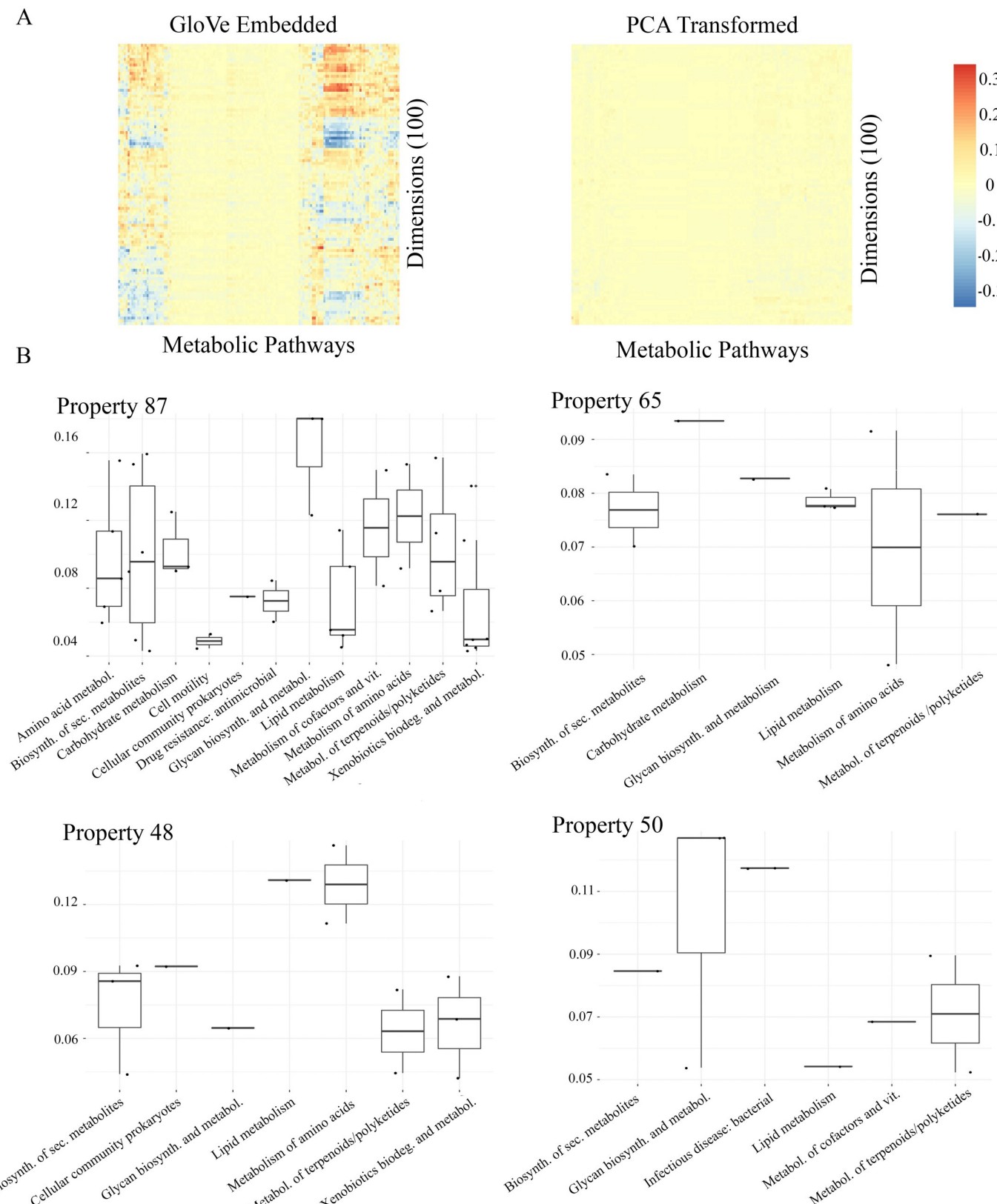

**Fig 7.** Dimensions in GloVe embedding space correlate with some metabolic pathway annotations, but dimensions in PCA embedding space do not (A). Each column in each heat map represents a metabolic pathway from KEGG (e.g. ko00983). Each row is a dimension in either GloVe or PCA embedding space. Significant metabolic pathway correlations of the four properties strongly associated with IBD in both Halfvarson and AGP datasets (B). Each point represents a metabolic pathway, the x axis shows the pathway's broad category, and the y axis shows the strength of the correlation.

the possibility of a connection between annotated genetic capacity to express metabolic pathways and the properties that make up embedding space. First, we find each Amplicon Sequence Variant's (ASV) nearest neighbor in the KEGG database [41] using Piphillan [42], and use the KEGGREST API [43] to determine which pathways are present in that ASV's genome. This results in an ASV by pathway table where there are 11,893 ASVs with near neighbors in the database, and 148 possible metabolic pathways. Then, we identify the significantly correlated metabolic pathways for each property in embedding space. A permutation test is used to simulate a null distribution of maximum correlations per embedding property and determine significance. We find that every property significantly correlates with at least 1 annotated metabolic pathway. S1 Table shows each dimension and its significantly correlated metabolic pathways; each dimension has significant correlation with 3 to 57 pathways. We see that the magnitude of correlations between embedding dimensions and metabolic pathways are far greater in the GloVe embedding case than in the PCA-transformed case (Fig 7A). Additionally, none of the correlations between PCA dimensions and metabolic pathways are significant under a permutation test after multiple hypothesis correction (S4 Fig). This suggests that the properties learned by the GloVe algorithm based on co-occurrence patterns between ASVs may actually reflect the metabolic capacity of taxa.

In addition, we calculated distance between properties based on both their ASV vectors, and their correlation vectors with pathways. This results in two property by property dissimilarity matrices using the two definitions of properties (ASVs vs. pathways). We find that properties that are similar to each other based on their ASV vectors are also close to each other based on their correlations with annotated metabolic pathways (Mantel p = 9.99 $^*$ 10^-5, r = 0.38).

## Interpreting the predictive model for IBD with metabolic pathways

In order to explore the implications of properties and metabolic pathways for IBD, we calculated an association score (see method section "Importance of properties and pathways in predictive model") between each property and a positive IBD prediction. The full tables of the most predictive dimensions, their associated metabolic pathways, and their direction of influence on the prediction can be found in S2 Table (AG) and S3 Table (Halfvarson). First, we identified those properties strongly associated with a positive IBD prediction (association score above 8 in both Halfvarson and AGP datasets). We then selected all metabolic pathways significantly correlated with more than one of those highly relevant properties. In this way, we identified 45 metabolic pathways of interest for IBD (S4 Table). The pathways fall broadly into 10 main categories according to the KEGG Brite database: carbohydrate metabolism, lipid metabolism (including steroid metabolism), metabolism of terpenoids and polyketides, glycan biosynthesis and metabolism, amino acid metabolism, metabolism of cofactors and vitamins, antimicrobial drug resistance, bacterial pathogenic markers, cell motility and cellular community formation, and xenobiotics biodegradation/other metabolic function.

## Explaining the variance in properties

Lastly, we sought to determine how much of the information contained in properties can be recapitulated by looking at the above described annotated metabolic pathways, and how much was unique to each property. For each property, we use a linear regression to predict the

property values per ASV from the pathway presence/absence per ASV. We report the $r^2$ statistic per property, and find that metabolic pathways can explain a maximum of 36% of the variance in one property, and a minimum of 11% of another. This means that, while there is a strong correlation between properties and annotated metabolic pathways, most of the information contained in properties are not represented by annotated information (S5 Fig)

## Discussion

In a data-driven field dominated by small sample sizes and large variable spaces, it is necessary to employ some form of dimensionality reduction. Currently, this is done by filtering by taxa prevalence, clustering based on phylogenetic proximity, or is not done at all. We present here a method to leverage massive public datasets to learn an embedding space that represents the latent properties driving taxonomic abundances.

We demonstrate that we can learn a fecal microbiome property space that is more apt at predicting the IBD status of the host than non-reduced and pca-reduced spaces, and that enables more robust generalization of prediction models between datasets. We lastly define the relationships between embedding space and known metrics used to explore microbiomes like phylogenetic distance and metabolic pathway genetic capacity.

### Properties

We have shown that embedding an ASV table into property space using GloVe integrates patterns from public data into modeling efforts, producing more accurate diagnostics while decreasing data dimensionality. Classifiers built after transforming data in this way are more robust, and the same embeddings generalize to improve the accuracy of classifiers built from completely independent datasets. Properties also allow models trained on one dataset to be applied to another independent dataset with positive results.

In addition to improving classification accuracy for IBD, the embeddings quantify and simplify the microbial landscape of gut microbiomes. Rather than considering a microbiome as a collection of bacterial counts, we propose to describe a microbiome as a vector of values for the relevant properties. Because these properties are learned from the data directly, they are much less biased than manually engineered features. Analysis performed on this latent property space is likely to be much more robust to variations in datasets than that based on taxon counts.

### Biologically driven dimensional reduction

We use unsupervised learning to define an embedding space where taxon proximity represents similarity in co-occurrence patterns. Unsupervised learning limits the human decision-making bias in property definition, but also produces unlabeled properties whose interpretation is not immediately obvious. We hypothesize that co-occurrence patterns are driven by taxon function like metabolism, synthesis of secondary metabolites, and secretion of antimicrobial products. We show in our pathway analysis that property distributions in fact do correlate significantly with metabolic pathways, therefore, the learned property space is likely informed by taxon function from within a biological context. Some elements of property space may also be informed by other factors, such as geography or diet commonalities between groups of people, and this should be explored further.

### Annotation independent

While we have explored the associations between embedding properties and the annotated quantities of genetic potential and phylogeny, the power of this embedding technique is that it

does not rely on annotations of known taxonomic groupings or full genomes in order to improve prediction accuracy of host phenotype. Because any ASV that has been observed during embedding training can be embedded, it is possible to describe the properties of uncultured and unannotated ASVs, and include this information in a classifier. The transformation into embedding space requires only an ASV table, and uses no sample associated data like lifestyle variables or diagnoses.

## Relationship to phylogeny

Results from section "Property relationship with phylogeny" suggest that phylogeny does drive property structure to some extent, as might be expected. Using embeddings, we identified 3 genera, *Streptococcus*, *Staphylococcus*, and *Neisseria*, that may interact synergistically as their members exhibit similar patterns to each other in embedding space and also frequently co-occur directly. Literature does not suggest that the most heavily studied members of these genera, the pathogens *S. aureus*, *S. pneumoniae*, *and N. meningitidis*, directly interact, but little is known about the other members of these genera. We also identified 3 pairs of ASVs that are close in embedding space but *do not* co-occur directly; we hypothesize these microbes may therefore share similar ecological functions. The first pair is annotated as *Bacteroides vulgatus* and *Escherichia coli*. Both these species are known to cause gastrointestinal inflammation and colitis, and have not been reported in the literature to depend on one another, suggesting these microbes may fill the same niche in the human gut [44]. The second pair is *Bacteroides uniformis* and *Faecalibacterium prausnitzii*. These species are on the short list of 'next generation' probiotics that yielded positive outcomes for inflammatory and metabolic disorders, potentially implying their similar functions [45,46]. The third pair is *Faecalibacterium prausnitzii* and *Eubacterium rectale*. These species are heavy butyrate producers, making it possible that they perform similar ecological functions in the human gut [47].

## Implications for IBD

We were able to identify 10 main categories of KEGG BRITE pathways that were significantly correlated with properties associated with IBD (S4 Table). Among these pathways, both steroid metabolism and biosynthesis were found to be associated with IBD. Steroids are a well-known and commonly utilized treatment for patients with active Crohn's disease [48]. Enrichment in steroid metabolism in the gut microbiome could be reflective of an increase in steroid availability due to treatment.

Several pathways belonging to the rather broad BRITE category of "other metabolic function" have already been well explored and characterized in the literature as related to IBD. Toluene degradation (KEGG pathway 00623) was found to be increased in both Crohn's disease (CD) and Ulcerative Colitis (UC) samples in a microbiome survey meta-analysis [49]. Components of the benzoate metabolic pathway, including fluorobenzoate degradation (KEGG pathway 00364), were associated with IBD severity in a treatment-naive cohort with CD [50]. Analysis of inflamed gut lining mucosa in patients with IBD also found decreased ascorbic acid content (KEGG pathway 00053)[51] All of these pathways, along with dioxin degradation, inositol phosphate metabolism, and lipoic acid metabolism, were associated with an IBD prediction in our model.

We also found multiple glycan biosynthesis pathways correlated with predictive IBD properties (KEGG pathways 00511, 00514, 00515, 00601). In particular, bacterial glycosphingolipid biosynthesis, a pathway which has anti-inflammatory effects when produced by the host epithelial cells [52], was found to be associated with IBD in our model. We speculate that this may indicate a shortage of glycosphingolipids in the gut environment, exacting positive selection

pressure on microbes that can produce their own. Lipopolysaccharide biosynthesis and multiple types of O-glycan biosynthesis were also implicated in our model, all of whose association with IBD has been explored, briefly, in the literature [53–55]. Given its importance and consistency in our predictive model, this group of pathways may warrant further exploration.

## Limitations and future expansion of the work

While embedding Amplicon Sequence Variants (ASVs) affords the benefits to classification and interpretation previously discussed, it relies heavily on the definition of a "biologically meaningful unit" which will then be embedded. For the sake of between-study replicability, we choose to measure the co-occurrence patterns of ASVs [35] as a base unit. It may, however, prove more informative to define a biologically meaningful unit in another way. For example, perhaps ASVs clustered at a 99% threshold more accurately capture meaningful patterns in co-occurrence. We may also consider a variable threshold that is more representative of common ancestry on a phylogenetic tree and aggregate based on clade architecture before embedding.

Additionally, the presented set of embeddings was constructed using only the forward reads from the American Gut dataset, as reverse reads were not provided in the EBI database. Future embeddings constructed from full length V4 or other 16S hypervariable regions will likely provide more accuracy and specificity. New embedding transformation matrices would need to be trained for each new biome or segment of 16S gene being explored.

In its current form, the algorithm does not make specific considerations for differences in sequencing depth, which affects how many taxa can be observed in a given sample. Future iterations of this method could include weights such that the observed absence of taxa in a sample with a large number of reads is weighted more heavily than the absence of taxa in a sample with fewer reads.

While the construction of embeddings is not affected by the inconsistency of self-report data, the accuracy of the classifier may be. In this study, we considered only a self-reported medical professional diagnosis to be accurate, and rejected any self-diagnosis reports. While it is possible that classifier performance would change with the inclusion of more liberal diagnostic criteria, the strict diagnosis definition successfully generalized to an independent dataset, which was not self-reported [8].

Properties in embedding space have strong associations with metabolic pathway potentials, but it remains unclear whether they truly represent the expression of those pathways. Future development could also consist of integrating multi-omics datasets available in other studies, including the Human Microbiome Project. Wet lab validation of these hypothesized property-metabolic expression associations would verify the ability of GloVe embeddings to predict metabolic expression from 16S data. This would allow for the integration of metabolic data from all observed taxa, not just those few whose full genomes are available in databases.

It might be possible to use the embedding space to identify taxa that form stable communities together—taxa that are close in embedding space may stabilize each other in culture and *in vivo*. Through mechanisms like cross-feeding, joint nutrient acquisition, and other cooperative behaviors, microbes may form groups that are more versatile and secure than the individual species on their own. Taxa near each other in embedding space, if they are not directly interacting, may have synonymous functions in their respective communities. By clustering and categorizing microbes by their respective roles, we may gain insight into which bacterial populations secure one another's stability. Particularly, the relationship between phylogenetic distance and distance in embedding space may be of interest. Microbes that are very closely related to one another through evolution but have very dissimilar co-occurrence patterns may be particularly predictive of their environment, as they have specialized quickly and efficiently.

It may be that different variable regions better capture the co-occurrence patterns of taxa, and so are more representative of taxonomic relationship to the environment.

Lastly, the embedding framework can be applied to any system or base unit of interest. It may be particularly illustrative to embed genes from metagenomic datasets instead of taxa. This would allow us to determine mathematical representations of the context of each gene, as well as to glean the robustness and reproducibility benefits from dimensionality reduction for metagenomic data. As always, appropriate benchmarking and exploratory analyses will be necessary to determine the appropriate use cases for this technology.

## Conclusion

By integrating patterns from public datasets into individual survey studies, we bring the increased statistical power and generalizability of results of meta-analyses into each independent study. While this work shows the value of an embedding framework for predicting IBD from the gut microbiome, this same framework can be leveraged in any environment with enough data and for any predictive problem of interest. Furthermore, we assert that analyses that define microbiomes by their latent properties instead of by their taxon member list may offer more information, more reproducibly, and more relevance to the macroscopic world.

## Materials and methods

Code available at: https://github.com/MaudeDavidLab/gut_microbiome_embeddings.
Data files available at: http://files.cgrb.oregonstate.edu/David_Lab/microbiome_embeddings/

## Workflow

The workflow is as described in Fig 1:

First, we learn the embedding space using ASV-ASV co-occurrence data from the American Gut Project (A). The data contains 18,480 samples and 26,726 ASVs. Two ASVs are considered co-occurring if they are detected in the same fecal sample. From the patterns of co-occurrence across all samples, the GloVe algorithm produces a transformation matrix, where each ASV is represented by a vector in embedding space (B). We call each dimension in embedding space a "property" ($P\_1 \ldots P\_k$) as each is a set of numbers used to differentiate ASVs' co-occurrence patterns. No metadata is used to create the embeddings; the process is completely unsupervised.

To transform the dataset of interest into embedding space (E), we take the dot product between the dataset (D) and the transformation matrix (C). The dot product operation outputs a matrix of samples by properties, where property vectors are calculated as the weighted sum of property vectors over all the ASVs present in that sample.

Lastly, we input the transformed data into a random forest classifier (F), along with 13 sample-associated features like exercise frequency, probiotic consumption, frequency of vegetable intake (G), to train a model that predicts IBD vs. Healthy host status. Samples and their associated features can be found in S5 Table.

In total, three random forest classifiers are trained, with the three types of input data: GloVe embedded, PCA transformed, and non-embedded normalized count data. Each classifier was cross-trained on 85% of samples to optimize hyperparameter choices for the number of decision trees, the depth of each tree, and the weight put on a positive classification.

## Embeddings

There exist some easily drawn parallels between natural language data and microbiome data, namely that documents are equivalent to biological samples, words to taxa, and topics to

microbial neighborhoods [30]. Just as a book may be defined by the aggregate topics it discusses, a microbial environment may be defined the neighborhoods or communities it contains.

In this study, we take advantage of another connection between words and microbes, that is the capacity of both entities to be described by a finite set of discrete, characteristic properties. For instance, the word 'apple' in English can be defined as an edible, red, non-gendered, crunchy, object. Similarly, the species *Clostridium difficile* can be defined as a spore-forming, infectious, spindle-shaped bacteria. While it would be difficult to distinguish between a recipe book and a magazine of food reviews by enumerating differences in the occurrence of individual words, differentiating the two becomes simple if we select appropriate properties. While both media use words that have high scores in the property "edibility", the recipe book also uses words that have a high declarative score, like 'cut', 'wash', and 'prepare', while the food review uses words that have high descriptive scores, like 'fantastic', 'delectable', or 'abysmal'. Just as the properties of "declarative" and "descriptive" allow us to differentiate texts more effectively, property-based analysis of microbiomes allow us to distinguish between two microbial scenarios more easily than individual taxon counts. Analysis on the level of properties thus provides a more accurate and generalizable representation of the data's structure.

Embedding is a technique used ubiquitously in machine learning, especially in natural language processing [56–58]. Embedding algorithms take discrete units of data (e.g. words or taxa) and embed them into a vector space, preserving proximity between the units based on any metric that can compare two units. In the case of embedding taxa, possible metrics include phylogenetic distance, genome similarity, or morphology: in this paper the chosen proximity metric between units is patterns of co-occurrence. The embedding algorithm used in this paper is GloVe, an algorithm designed for word processing [59]. Using this algorithm, two taxa that occur with similar sets of other taxa at similar frequencies should be close in embedding space, and two taxa that are found in the presence of different neighbor sets should be far from each other. To visualize this, we return to the analogy of word analysis. Two words, "apple" and "banana", are close to each other in embedding space because they tend to occur with similar sets of words like "eat", "fruit", "tasty", and "smoothie". Likewise, the words "king" and "marshmallow" tend to occur in different contexts; "king" is most often found in the company of words like "politics", "throne", and "empire" while "marshmallow" is found with words like "toddler", "fluffy", and "scrumptious". Note that there are two ways words may be close in embedding space. First, words may directly co-occur frequently, like the words "apple" and "banana". Instead, words may be synonyms, which do not often co-occur directly with each other, but instead co-occur with similar patterns, like "large" and "huge" both being used to describe giants, mountains, and appetites. Returning to the concept of embedding taxa, we may use embeddings to discover relationships both between taxa that work together directly, and between taxa that are synonymous and likely fill the same niche.

Once proximity in embedding space has been established, the data can immediately be used to improve modeling efforts. Subsequently, conceptual properties can be assigned to the learned dimensions by observing which entities have similar values in any given dimension. If "strawberry", "cookies", "cake", and "ice-cream" all have high values in one dimension, and "mud", "medicine", and "brussel sprouts" all have low values in that same dimension, we may call that dimension the "delicious" property.

## GloVe algorithm

We used the GloVe algorithm [59] on ASVs to generate embeddings. Briefly, the embedding algorithm (Fig 1B) learns ASV representations that maintain co-occurrence patterns between

pairs of ASVs. In this algorithm, the metric to be preserved is a function of $\frac{P_{ik}}{P_{jk}}$, the ratio of probabilities of co-occurrence between ASV i and ASV j with k, respectively. Variables i and j are the ASVs being related, and k is a third context ASV. The algorithm learns word vector representations using a linear regression such that the dot product of any pair of word vectors is proportional to the probability of their co-occurrence. The probability of co-occurrence between any two ASVs i and j is calculated as the sum of all co-occurrences between i and j divided by the number of occurrences of i. ASVs were considered co-occurring if they occurred in the same sample.

The result from this algorithm is a representation of each ASV in x-dimensional space, where x is chosen by the user. The x-dimensional space is shared across all ASVs, and thus each dimension can be interpreted as a property for which each ASV has a value. The number of dimensions, x, is a hyperparameter to be tuned, and results are reported for a range of dimensions: 50, 100, 250, 500, and 750. Embeddings were learned on 85% of the data, which 15% of samples set aside for testing.

## Data transformations

**Normalization.** The inverse hyperbolic sin function, $sin^{-1}(x) = log(x+(x^2+1)^{1/2})$, was selected because it mimics the function $log(2x)$ function almost exactly, except for behavior near 0 [60]. Below 1, log will return a negative value, and at 0, log is undefined. In contrast, inverse hyperbolic sin does not fall below 0 when the argument is low, and is defined as 0 at 0.

**Embedding space.** We transform samples into embedding space by taking the dot product between the query sample by ASV table and the taxon by property embedding transformation matrix. A sample's value for a property 1 is $\sum_{i=1}^{k} T_i * P_{T_i 1}$ where T is the sample's vector of ASV abundances, P is the ASV's vector of property values, and k is the number of ASVs in the sample. To perform this operation, ASVs in the query set (Fig 1D) must match those in the embedding transformation matrix (Fig 1C) exactly. To allow for the fact that ASVs in one study may exhibit variation such as a difference in length, we consider ASVs to match between datasets if they are at least 99% similar and align with an e value less than 10^-29, as determined using BLAST [61]. If an ASV from the query set does not match with these criteria to any ASV in the embedding transformation matrix, it is dropped.

**PCA transformation.** Data is first normalized using the inverse sin hyperbolic function, and then transformed using Principal Coordinate Analysis (PCA). PCA is an ordination technique that projects samples into lower dimensional space while maximizing the variance of the projected data [62].

## Random forest predictions

The value of the embeddings was evaluated by success at predicting host IBD status using a random forest model [62]. The model was built using Python sci-kit-learn, and hyperparameters for the depth of tree, number of trees, and weight on a positive prediction were selected using 10-fold cross validation on the training set. A different model with different hyperparameters was built for each data type: normalized taxonomic abundances, PCA transformed normalized abundances, and GloVe embedded normalized abundances. Models also included self-reported sample metadata such as exercise, sex, daily water consumption, probiotic consumption, and dietary habits when trained on the American Gut data. Models were evaluated by their performance, namely area under the receiver operating curve, on the held out test set of 15% of samples.

## Correlations with KEGG pathways

For each ASV, we find its closest match, thresholded at 97% similarity, in the KEGG database using the software Piphillan [42]. Each possible metabolic pathway then gets assigned a 0 if it is absent or a 1 if it is present in that nearest neighbor's genome.

Limiting the following analysis to include only those ASVs that had near neighbors in the database, for each of the properties in embedding space, we find its maximally correlated (absolute value) metabolic pathway. Then, to ascertain whether those correlations were significant, we applied a permutation test [63]. We constructed 10,000 null pathway tables by permuting the rows of the original pathway table. We repeated the above procedure, finding the maximally correlated pathway for each of the embedding dimensions in each of the null pathway tables. This results in 10,000 maximum correlation values per embedding dimension, which form a null distribution for each embedding dimension. The significance of the statistic in a permutation test is calculated as the number of times a maximum correlation in a null pathway table was more extreme or equal to the maximum correlation actually observed. Dimensions (columns) in both GloVe transformation matrix and PCA rotation matrix space are normalized to mean 0 and variance 1 to account for differences in scales between the two spaces. We report both the maximally correlated pathway for each property, all of which are significant, and also *all* significantly correlated pathways per property.

## Calculating phylogenetic distances

We produced a Multiple Sequence Alignment and subsequently a phylogenetic tree of all ASVs, using Clustal W2 [64] multiple alignment and phylogeny creation software. The tip-to-tip phylogenetic distances were then calculated between every pair of ASVs using the dendropy python package [65].

## Explaining variance of properties with metabolic pathways

For each property, we set up a linear regression where the property values per ASV are the response variable, and the pathway presence/absence for each ASV are the independent variables. The $r^2$ statistic is reported to assess the variance in property values explainable by the presence of annotated metabolic pathways.

## Importance of properties and pathways in predictive model

In order to calculate the direction of association of a property with disease, we limit each tree in the random forest to split on 3 variables. We then backtrace; if a higher value of the property led to an IBD prediction, we add one to the association score between IBD and that variable. Likewise, if a lower value of the property led to IBD, we subtract one from the association score.

In calculating metabolic pathway importance to the predictive model, we first find all properties that are consistently associated with health or with IBD. Then, we count the number of times each pathway is significantly correlated with one of those properties. If a pathway is significantly correlated with more than two consistently predictive properties, it is considered important in that phenotype.

## Dataset

Embeddings were trained using data from the American Gut Project [24]. This crowdsourced project provides 16S samples from the United States, United Kingdom, and Australia, along with associated dietary, lifestyle, and disease diagnosis information. Amplicon Sequences

Variants (ASVs) were called using the DADA2 algorithm [66], resulting in 18,750 samples and 335,457 ASVs. Samples with fewer than 5,000 reads and ASV's not occurring in at least .07% of samples (13 samples) were then discarded, resulting in 18,480 samples and 26,726 ASVs. Embeddings were trained on a randomly selected 85% of the filtered samples, and the other 15% were set aside for classifier testing.

Training embeddings does not require labeled data, and so samples could be used irrespective of their available metadata. The machine learning classifier was trained and tested only on samples that had a positive or negative IBD diagnosis, 5018 and 856 samples respectively. IBD diagnosis was provided in various self-reported options from the American Gut study: "I do not have this condition", "Self-diagnosed", "Diagnosed by a medical professional (doctor, physician assistant)", or "diagnosed by an alternative medicine practitioner". For this study, we considered only samples claiming a medical professional diagnosis to be true.

Lastly, in order to test the generalizability of embeddings, we used 16S data on patients with Crohn's Disease (CD) and Ulcerative Colitis (UC) and healthy controls from Halfvarson et al. [8]. DADA2 [66] was again used to call ASVs, samples were discarded if they had fewer than 10,000 reads, and ASVs were not filtered for prevalence. After quality control, 26,251 ASVs remain, 17,775 of which have near neighbor representations in embedding space. The dataset included samples with multiple diagnoses, but for the sake of consistency, we focused on the most common diagnoses of Crohn's disease, Ulcerative Colitis, and healthy control. In total, this left 564 samples from 118 patients, as the dataset contains multiple timepoints for each patient. When models were trained and tested on Halfvarson datasets, timepoints from the same patients were included entirely in the train or test set, so as not to train then test on the samples from the same patient. The same pipeline was used to process the Schirmer et al. dataset [9] resulting in 197 samples and 5869 ASVs, 3135 of which has near neighbor representations in embedding space using a 99% similarity threshold.

## Software and packages

**Python Packages:** Pandas 0.23.4, Numpy 1.16.3, Sklearn 0.20.2, Scipy 1.2.0, Matplotlib 3.0.0, Re 2.2.1, Skbio 0.5.5 **R packages:** pheatmap_1.0.12, cowplot_0.9.4, ggplot2_3.2.0, RColorBrewer_1.1–2, Gtools_3.8.1, Dada2_1.10.1, Rcpp_1.0.1, plyr_1.8.4, stylo_0.6.9, KEGGREST_1.22.0

## Supporting information

**S1 Table. Property-pathway correlations.** Properties matched to their maximally correlated metabolic pathway. Includes KEGG pathway identifier and annotated pathway name. (TXT)

**S2 Table. Numeric metadata.** Samples and sample associated information converted into numeric quantities for machine learning. (CSV)

**S3 Table. Property importance in IBD AG dataset.** Properties listed by importance in differentiating between IBD and healthy control samples in American Gut data using a random forest with a depth of 2. Each property is labeled by its maximally correlated metabolic pathway, and the direction of association it has with disease. The last column reports the cumulative number of trees in a cross-validated random forest that support that association. (CSV)

**S4 Table. Property importance in IBD Halfvarson dataset.** Properties listed by importance in differentiating between IBD and healthy control samples in Halfvarson data using a random

forest with a depth of 2. Each property is labeled by its maximally correlated metabolic pathway, and the direction of association it has with disease. The last column reports the cumulative number of trees in a cross-validated random forest that support that association.
(CSV)

**S5 Table. Pathway importance in IBD.** List of pathways significantly correlated with properties strongly associated with IBD.
(CSV)

**S1 Fig. AG predictions with different ASV matching thresholds.** Embeddings were trained on American Gut data, and the predictive models were trained and tested on Halfvarson dataset. Transforming microbiome data into GloVe embedding space (blue, 100 features) prior to model training produces more accurate models than using ASVs (purple, 26,251 features). ASVs from the either dataset were matched to the embedding transformation matrix if sequences were 100% (A) or 97% (B) similar.
(TIF)

**S2 Fig. Other predictions with different ASV matching thresholds.** Two models, one embedding-based and one ASV-based, where trained on American Gut data and tested on Halfvarson dataset (A) and Schirmer dataset (B). Embedding-based models outperform ASV-based models significantly. ASVs were matched to the embedding transformation matrix if sequences were 100% or 97% similar, as indicated.
(TIF)

**S3 Fig. Cooccurrence of Staphylococcus, Neisseria, and Streptococcus genera.** PCoA ordination of randomly selected 5000 ASVs using correlation of normalized counts. 3 genera, Staphylococcus, Neisseria, and Streptococcus that have similar embedding patterns likely co-occur directly.
(TIF)

**S4 Fig. Null property-pathway correlations.** Heatmaps showing correlations between dimensions in transformed space and one null annotated metabolic pathway table. We see far fewer and less dramatic correlations between transformed data and metabolic pathways when the pathway table has been shuffled.
(TIF)

**S5 Fig. Percent variance of properties explained by pathways.** Histogram depicting the percent variance of properties explainable by annotated metabolic pathways. Most properties are less than 25% explained by pathways, and no property is more than 36% explained.
(TIF)

## Acknowledgments

Thanks to the Piphillan Development Team at Second Genome, for running the appropriate version of Piphillan to align ASVs to the KEGG Database and to Nathan Waugh for manuscript editing.

## Author Contributions

**Conceptualization:** Christine A. Tataru.

**Data curation:** Christine A. Tataru.

**Formal analysis:** Christine A. Tataru.

**Funding acquisition:** Maude M. David.

**Investigation:** Christine A. Tataru, Maude M. David.

**Methodology:** Christine A. Tataru, Maude M. David.

**Project administration:** Christine A. Tataru.

**Software:** Christine A. Tataru.

**Supervision:** Maude M. David.

**Validation:** Christine A. Tataru.

**Visualization:** Christine A. Tataru.

**Writing – original draft:** Christine A. Tataru.

**Writing – review & editing:** Christine A. Tataru, Maude M. David.

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
