## [Decision Letter · Decision Letter 0]

24 Oct 2019

Dear Dr Tataru,

Thank you very much for submitting your manuscript 'Decoding the Language of Microbiomes using Word-Embedding Techniques, and Applications in Inflammatory Bowel Disease' for review by PLOS Computational Biology. Your manuscript has been fully evaluated by the PLOS Computational Biology editorial team and in this case also by independent peer reviewers. The reviewers appreciated the attention to an important problem, but raised some substantial concerns about the manuscript as it currently stands. While your manuscript cannot be accepted in its present form, we are willing to consider a revised version in which the issues raised by the reviewers have been adequately addressed. We cannot, of course, promise publication at that time.

Sincerely,

Jacopo Grilli

Associate Editor

PLOS Computational Biology

Stefano Allesina

Deputy Editor

PLOS Computational Biology

[LINK]

Both the reviewers expressed interested in the paper, but the also raised major points regarding the presentation and the interpretation of results.

Reviewer's Responses to Questions

**Comments to the Authors:**

Reviewer #1: In this manuscript, the authors proposed and evaluated an embedding methodology for taxa. In detail, the proposed method embeds taxa into meaningful numerical vectors using AGP co-occurrence table. Then the authors demonstrate that this embedding can be used for other datasets and produce better and more robust results. One of the advantages of their embedding the authors claimed is that the embedding essentially encodes the properties of those taxa and thus are more informative. Authors compared their proposed model with other methods. They showed the improvement of classification accomplished by their method, and the results indicate a possibility of being biologically meaningful by correlating to metabolic pathways. However, the interpretation of the co-occurences and properties is not investigated fully and could be developed more.

MAJOR Comments:

* The biggest aspect from this paper is that the method is learning from all the co-occurrence patterns but never really revealing the learned patterns. There is much interest in knowing which species are actually co-occurring: https://journals.plos.org/ploscompbiol/article?id=10.1371/journal.pcbi.1002606

Using the OTU probabilities and ratios, can the authors shed light on common patterns to certain sample-associated metadata?

* Fig. 6 is highly confusing, and Section 2.6 can be improved. The authors plot the cosine distances between embedded-vector OTUs vs. their tip-to-tip branch phylogenetic distance. If well correlated, one would see that as the cosine distance increased, the phylogenetic distance would increase. However, this is not the case. We see two islands and not much explanation about why this is in Section 2.6. Why not plot the actual OTU embeddings – could use T-SNE for visualization? Then, it is possible to visualize the distances in the embedding space. The OTUs could then be color-coded via their genus/species/order/phylum level labels to show taxonomic clusterings.

* Another idea to remedy Fig. 6 is to hierarchically cluster the OTU embedddings and see how well that tree reconciles (similarity metrics between the two trees) with the phylogenetic tree. IE: there are many ways to better test the idea in Fig. 6.

* Section 2.7 is a good way to interpret the properties. Since the authors claims that the properties are actually related to the pathways, more detailed analysis and visualization can help here. Perhaps T-SNE be used to plot/cluster the properties (by using vectors of their correlations to pathways)? Can we see if properties that are similar, have similar metabolic pathways? Can we see if properties that are similar, are more similar taxonomically. The properties might be learning a group of taxa as well.

* Figure 1 and section 4.1 is not clear about how the co-occurrence table is made from the ASV table. I think this algorithm should be described in the paper as part of your pipeline. E.g., what is the dimension of the co-occurrence table? How are the values determined? How is the probability of co-occurrence of taxa i and j with k calculated? Only one sentence (in line 604) talked about the co-occurrence but didn’t mention the computation of the probability of co-occurrence.

* Figure 1 and section 4.2 is confusing as there is no 3-dimensional tensor at all in the process. In figure 1, it is correct to have D * C = E. But the intermediate 3D tensor should be removed. The authors can simply describe the averaging idea in 4.2. It looks like D * C becomes a 3-d tensor which is confusing. Also, when applied to a new dataset, what if the dimensions of D and C don’t match? Do you drop the columns in C or use the average for those columns?

* The use of ASVs is very confusing. ASV’s are meant to improve on OTU thresholding. However, the authors say that they consider any two taxa the same if they have more than 97% identity. This defeats the purpose of ASVs altogether! Since the authors use dada2 pipeline, why not use its results…. Why then use 97% threshold on top of that? The authors need to compare this to using actual ASVs that are not grouped together. Is performance worse when using pure ASVs?

* Can you sweep the dimensions and find a similar plot to Fig. 2 when using actual ASVs and not those thresholded to 97%?

* The author mentioned that the counts were normalized by an inverse hyperbolic sin function.

Is this for raw count input data only (since this sentence is right after line 523 where the authors listed all three possible input formats to the RF model)? If so, why not relative abundance or maybe centered log ratio transform?

MINOR Comments:

* In Figure 3B2 and Figure3B3, the authors have GloVe embedding data and PCA data in 113 dimensions (which after reading the methods is the 100 properties plus 13 sample-associated metadata features). This is consistent with the text. However, for Non-reduced data, in Fig 3B1, it says that 26726 features are used but in the text, it says that 26739 features are used. What is true here? Were 26726 or 26739 features used for the non-reduced RF model for training?

* The transition between first several sections in results section (2.1, 2.2, 2.3) seems not smoothing. Maybe in the Result section, first give an overview of the data and target (problem setup), then get to the specific results.

* Figure 7 is upside down.

* In line 497, the link to the github repo is 404 NOT FOUND. I only found https://github.com/MaudeDavidLab/embeddings_final (please update the link in your manuscript). The README could provide explicit examples instead of vague statements. Instead of “Run Piphillan”, can you give the exact command that was run? Can you give a toy example of a dataset that someone may want to run and then step-by-step exact commands that show how to run the whole pipeline?

* Organization of the paper can be improved. Ideas are provided in different sections in a fragmented way: First 2 paragraphs of section 3.1 are the explanation of the employed method. So, they can be moved to the Materials and Methods section. Third and fourth paragraphs of section 1.3 are also explaining the idea of the connection between words and how properties capture these connections. These paragraphs can be moved to the method section.

* Page 19, Section 4.1 line 502 and 503: “In this algorithm, the metric to be preserved is a function of P_ik / P_jk, the probabilities of co-occurrence of taxa i and j with k.” is ambiguous. The notation implies that P_ik is divided by P_jk.

* Figure 7: The Caption of Figure 7 says “Each column in each heat map represents a metabolic pathway from KEGG e.g.321 ko00983). Each row is a dimension in either GloVe or PCA embedding space”. This means that in the graph each point represents a correlation coefficient. However, in the figure, both labels are provided for the y-axis (on either side). Based on the caption, “metabolic pathway” is x-axis, not y-axis. Also, a legend that explains the numerical values of color codes can improve the figure.

* Figure 6: Caption of figure 6 says “lighter color signifies a higher density of taxa pairs” But the unit of density is not mentioned in the legend. Mentioning the unit will make the figure more specific and informative.

* Figure 4 B1, B2: Markers should be added to the plots

* In line 305, Piphillan is misspelled as Piphillian

Reviewer #2: This article describes a new method for making sense of microbiome data: using NLP technologies to learn the co-occurrence pattern of taxa, and convert the high-dimensional taxon table into a moderate-dimensional "property" table. Unsupervised models trained on this type of data were shown to be more accurate than those from original taxon counts or from PCA.

This method is interesting. It is distinct from traditional bioinformatic approaches used to analyze microbiome data. Although I am aware of technologies that involves taxon co-occurrence (e.g., SparCC, CoNet) (the authors didn't address them though), and efforts in analyzing microbiome data using NLP. Yet, the combination of them seems to be novel in this field. I think the method does have its value.

I also appreciate that the authors explored the biological meaning of the model. This is much better than those who just throw out an ML model and some AUROC plots without relating them to biology. Specifically, this article discussed the implication on phylogenetic relationships and metabolic pathways. I am bit surprised that "properties" were shown to be more informative than phylogenetic distance. I guess this may be because evolution AND ecology together shaped the niches of extant microbes. The tests on an IBD cohort is also valuable.

I also agree with some of the statements of the initiative of this study, such as line 87: "findings of differential abundance or function of a single species must be considered within its wider context of associated species and environmental factors."

However, there are still multiple limitations of this work. First of all, the authors made lots of hand-waving claims of superiority of this invention and how it will reframe "the microbiome analysis mindset" (line 49) or "shift the analysis paradigm" (line 173). I am highly skeptical about these statements. I suggest that the authors focus on science not marketing.

Second, the abstract and introduction are unnecessarily lengthy. This reduced my joy in reading this manuscript. Some paragraphs are just explanations of terms. For example, the whole section 2.1 is introducing AUROC and AUPR. I doubt if readers of PLOS Computational Biology would need this basic information.

What concerns me more is that the authors wrote long paragraphs reviewing (critizing) the current practices of microbiome studies. I understand that they wanted to highlight the value of this new method. But many points they made sound problematic to me. Examples:

Line 125: "several studies have attempted to integrate phylogenetic..." This sounds like few people are using phylogeny. But in contrast, technologies like UniFrac and Faith's PD have been massively used for more than a decade.

Line 89: "More specifically, predictive models that differentiate between disease and healthy guts based on microbiome composition in one dataset can rarely be successfully applied to samples from the same patient population collected independently." I doubt it.

The author's understanding of the "current" trend of differential abundance testing is outdated (e.g., citations 32 and 33). The authors repeatedly indicated that current researchers rely on the observation of differential abundance of individual taxa, rather than caring the overall image of the community. This is very exaggerated.

The authors indicated that abundance-based taxon filtering, taxon categorization and binning are forms of "dimensionality reduction" (line 106). I think most statisticians will not agree. Instead, what microbiologists usually refer to as "dimensionality reduction" include beta diversity distance matrices and multi-dimensional scaling techs based on them, such as PCoA and NMDS. I am surprised that these two terms were not mentioned at all in this manuscript, but instead the authors extensively discussed PCA. It was known that PCA on taxon counts does not yield as informative signals than PCoA on more deliberate beta diversity distance matrices.

Also for the author's information: Recent years have seen a new trend of differential abundance testing that is related to the authors' idea of using co-occurrence patterns. That is, to rely on the log-ratios among taxa instead of the relative abundance of individual taxa. I hope the authors can make some comments on those methods.

The test on Halfvarson data is great. But one single case study does not make a strong argument. Can the authors test some other datasets? Just a suggestion: Along with AGP, the Qiita platform hosts a large number of public 16S V4 datasets, and many of them were generated using a uniform protocol (the "EMP protocol"). This ensures low bias in cross comparison / meta analysis.

What is "normalized taxonomic count data"? The authors stated somewhere (line 526) that "Counts were normalized by applying an inverse hyperbolic sin function". This is not how microbiologists normalize their taxon tables, to my knowledge. Can the authors explain the rationale?

Fig. 3B2. It seems that taxon counts performs better than GloVe in AUPR. I appreciate that the authors are honest at this.

Line 289: Mantel r = 0.12 is quite low. The significant p-value is likely due to the large sample size (18,000). This section (2.6) overall does not sound like very strong evidence.

Finally, one trivial thing also affected my reading experience: The singular form of "taxa" is "TAXON"! Please correct it.

Final of finally, the source codes are available at GitHub, but they do not look like in the form that users can easily take advantage of. It seems to me that the authors simply dumped codes there to meet the journal's requirement. If the authors really plan to call for a paradigm shift in the field, they should spend some time on software engineering so that more people will come and experience this new method.

**Have all data underlying the figures and results presented in the manuscript been provided?**

Reviewer #1: Yes

Reviewer #2: Yes

PLOS authors have the option to publish the peer review history of their article (what does this mean?). If published, this will include your full peer review and any attached files.

Reviewer #1: No

Reviewer #2: No

---

## [Decision Letter · Decision Letter 1]

6 Mar 2020

Dear Ms. Tataru,

Thank you very much for submitting your manuscript "Decoding the Language of Microbiomes using Word-Embedding Techniques, and Applications in Inflammatory Bowel Disease" for consideration at PLOS Computational Biology.

As with all papers reviewed by the journal, your manuscript was reviewed by members of the editorial board and by several independent reviewers. In light of the reviews (below this email), we would like to invite the resubmission of a significantly-revised version that takes into account the reviewers' comments.

We cannot make any decision about publication until we have seen the revised manuscript and your response to the reviewers' comments. Your revised manuscript is also likely to be sent to reviewers for further evaluation.

Sincerely,

Jacopo Grilli

Associate Editor

PLOS Computational Biology

Stefano Allesina

Deputy Editor

PLOS Computational Biology

Reviewer's Responses to Questions

**Comments to the Authors:**

Reviewer #1: Major:

* I think the authors didn’t address our comments on finding species that are co-occurring. The authors complained that they can’t extract new information from their taxa-taxa co-occurrence network made by embedding vectors. But in fact, the authors can just do case studies in a few interesting species instead of plotting everything all at once in phylum level (which is actually very high level). For example, the authors can simple go to the OTU table and extract species that have high co-occurrence score (both have high abundance for different samples). Then just look at them. See you find species A and B are co-occurring and species C and D are co-occurring and species E and F are probably competing (one exist then the other doesn’t) Then in embedding space, you can plot the embedding of those species and see if you can discover the relationship in the embedding space. You might even be able to interpret some of the properties (dimensions in the embedded vector).

* As for t-SNE and PCA plots. Yes, if you are dealing a lot of taxa, it is very hard to visualize. But what if you just do a subset? For example, plot all the genera in a few families? Then we should expect that similar genera or families cluster together and distance from other families/genera?

Minor:

* Section 2.7 has been improved a lot. However, the procrustes analysis is not clearly explained in the method section in the paper. Like I am not certain what “ordinates properties based on their taxa vectors and pathway correlation vectors" -- what are the exact vectors being input into the Procrustes analysis? What do the long vectors out of the origin mean.. and why are they given numbers and not the shorter arrows outside? In the end, I’m not sure what this figure is showing.

* The authors use “et. al” and not “et al.”

* I would say that Supplementary Fig. 3 is still confusing and should be removed.

The rest of the paper seems fine:

* I think it is very helpful to show the tree made by embedding is significantly similar to phylogenetic tree. The author has addressed this issue.

* Figure 1 has been improved and the algorithm in 4.1 is clear now. In general, the paper’s method section is also clearer now.

*Clarifications on text and figures are noted.

Reviewer #2: The authors have addressed all concerns of mine. I have no more questions and recommend publication.

**Have all data underlying the figures and results presented in the manuscript been provided?**

Reviewer #1: Yes

Reviewer #2: Yes

PLOS authors have the option to publish the peer review history of their article (what does this mean?). If published, this will include your full peer review and any attached files.

Reviewer #1: No

Reviewer #2: No
---

## [Decision Letter · Decision Letter 2]

8 Apr 2020

Dear Ms. Tataru,

We are pleased to inform you that your manuscript 'Decoding the Language of Microbiomes using Word-Embedding Techniques, and Applications in Inflammatory Bowel Disease' has been provisionally accepted for publication in PLOS Computational Biology.

Best regards,

Jacopo Grilli

Associate Editor

PLOS Computational Biology

Stefano Allesina

Deputy Editor

PLOS Computational Biology

Reviewer's Responses to Questions

**Comments to the Authors:**

Reviewer #2: My concerns have already been addressed in the last round of review. I have no further questions and recommend for publication.

**Have all data underlying the figures and results presented in the manuscript been provided?**

Reviewer #2: Yes

PLOS authors have the option to publish the peer review history of their article (what does this mean?). If published, this will include your full peer review and any attached files.

Reviewer #2: No

---

## [Editor Report · Acceptance letter]

24 Apr 2020

PCOMPBIOL-D-19-01502R2 

Decoding the Language of Microbiomes using Word-Embedding Techniques, and Applications in Inflammatory Bowel Disease

Dear Dr Tataru,

I am pleased to inform you that your manuscript has been formally accepted for publication in PLOS Computational Biology. Your manuscript is now with our production department and you will be notified of the publication date in due course.

With kind regards,

Laura Mallard
